# Clinical and Genetic Characteristics of Korean Congenital Stationary Night Blindness Patients

**DOI:** 10.3390/genes12060789

**Published:** 2021-05-21

**Authors:** Hyeong-Min Kim, Kwangsic Joo, Jinu Han, Se-Joon Woo

**Affiliations:** 1Department of Ophthalmology, Seoul National University College of Medicine, Seoul National University Bundang Hospital, Seongnam 13620, Korea; hmkim3@gmail.com (H.-M.K.); joo_man@hanmail.net (K.J.); 2Institute of Vision Research, Department of Ophthalmology, Gangnam Severance Hospital, Yonsei University College of Medicine, Seoul 06273, Korea

**Keywords:** congenital stationary night blindness, gene, mutation, visual acuity

## Abstract

In this study, we investigated the clinical and genetic characteristics of 19 Korean patients with congenital stationary night blindness (CSNB) at two tertiary hospitals. Clinical evaluations, including fundus photography, spectral-domain optical coherence tomography, and electroretinography, were performed. Genetic analyses were conducted using targeted panel sequencing or whole exome sequencing. The median age was 5 (3–21) years at the initial examination, 2 (1–8) years at symptom onset, and 11 (5–28) years during the final visit. Genetic mutations were identified as *CNGB1* and *GNAT1* for the Riggs type (*n* = 2)*, TRPM1* and *NYX* for the complete type (*n* = 3), and *CACNA1F* (*n* = 14) for the incomplete type. Ten novel variants were identified, and best-corrected visual acuity (BCVA) and spherical equivalents (SE) were related to each type of CSNB. The Riggs and *TRPM1* complete types presented mild myopia and good BCVA without strabismus and nystagmus, whereas the *NYX* complete and incomplete types showed mixed SE and poor BCVA with strabismus and nystagmus. This is the first case series of Korean patients with CSNB, and further studies with a larger number of subjects should be conducted to correlate the clinical and genetic aspects of CSNB.

## 1. Introduction

Congenital stationary night blindness (CSNB) is a group of non-progressive inherited retinal diseases (IRDs) with dysfunction of rod photoreceptors or signal transduction between photoreceptor cells and bipolar cells, accompanied by various clinical features and diverse genetic mutations [1]. It has been discovered that defects in the visual signal pathway related to the rod photoreceptors, rod ON bipolar cell synapses, or retinoid recycling in the retinal pigment epithelium cause CSNB [2]. Visual symptoms primarily include night blindness with decreased visual acuity, refractive errors, nystagmus, or strabismus [3]. A recent study suggested that children with CSNB may present without complaints of night blindness [4]. In addition, these symptoms can overlap with other progressive IRDs, such as cone–rod dystrophies; thus, accurate diagnosis is essential to predict future visual outcomes. 

CSNB is categorized into four types according to electroretinography (ERG) and fundus abnormalities: Riggs type, Schubert–Bornschein type, fundus albipunctatus, and Oguchi [3]. Unlike fundus albipunctatus and Oguchi disease, the Riggs and Schubert–Bornschein types present normal fundus appearances, so they are frequently misdiagnosed as pathologic myopia or infantile nystagmus. The Riggs type shows a reduced wave in the dark-adapted (scotopic) response using ERG [5], whereas the Schubert–Bornschein type features a characteristic electronegative ERG pattern, detected as a normal A wave in the dark-adapted response with a drastically reduced B wave [6]. The Schubert–Bornschein type is divided into two subtypes: a complete type and an incomplete type [7]. 

The Riggs type genetic mutations have been previously reported, and the known pathogenic genes are autosomal dominant *GNAT1* (rod-transducing α subunit) [8,9,10], *PDE6B* (phosphodiesterase β subunit) [11,12], *RHO* (rhodopsin) [13,14,15,16], and autosomal recessive *SLC24A1* (sodium–calcium exchanger) [17]. Studies have found that Riggs type patients present relatively mild visual symptoms, such as restricted night blindness but normal photopic visual acuity, slight myopia, and no nystagmus [3]. The complete form of the Schubert–Bornschein type involves some notable pathogenic genes, such as X-linked recessive *NYX* (leucine-rich proteoglycan nyctalopin) [18,19,20], autosomal recessive *GRM6* [21,22], *TRPM1* [4,23,24,25,26,27,28,29,30], *GPR179* [31,32], and *LRIT3* [33], affecting signal transduction in the selective rod ON bipolar cell postsynaptic signal loss pathway. Typically, these complete type patients show moderate visual symptoms, including decreased visual acuity and high myopia. On the other hand, the incomplete form of the Schubert–Bornschein type is characterized by both ON and OFF response presynaptic signal dysfunction, and the light-adapted (photopic) responses are more severely reduced compared to the complete type. In addition, the light-adapted 30-Hz flicker response shows reduced amplitude with a double peak sign. The known pathogenic gene is X-linked recessive *CACNA1F* (calcium-channel alpha1 subunit) [3,7,34,35,36,37,38], and incomplete type patients present with varying degrees of visual symptoms, but commonly worse daylight symptoms than those with the complete type. 

To our knowledge, this study, for the first time, documents a case series of 19 Korean CSNB patients using clinical observations and genetic analyses. In Korea, CSNB-related studies with a large number of subjects and investigations of both clinical and genetic aspects have not been performed. Hence, we described the specific details of each case in this paper and were therefore able to comprehend the overall characteristics and causative gene mutations in Korean CSNB patients.

## 2. Materials and Methods

### 2.1. Patients and Clinical Data Collection

We enrolled 19 Korean patients with CSNB who visited two tertiary hospitals, Seoul National University Bundang Hospital and Gangnam Severance Hospital, between January 2009 and December 2018 (a 10-year period), and the final follow-up was performed until December 2020. This study was approved by the Institutional Review Board (IRB) of Seoul National University Bundang Hospital (IRB No. B-2101/663-102) and adhered to the tenets of the Declaration of Helsinki. Informed consent was obtained from all patients prior to the genetic analyses.

All subjects underwent full ophthalmic examinations, including best-corrected visual acuity (BCVA), fundus photography, spectral-domain optical coherence tomography (SD-OCT; Spectralis OCT; Heidelberg Engineering, Heidelberg, Germany), Goldmann perimetry, full-field standard ERG, and multifocal ERG. The type and frequency of the nystagmus was determined by visual inspection. Full-field ERG was performed using procedures based on the International Society for Clinical Electrophysiology of Vision (ISCEV) [39]. 

### 2.2. Genetic Analyses

A comprehensive custom gene panel of 295 known and candidate genes or a 429-gene targeted panel linked to IRDs was performed for genetic analyses as previously described in our reports [40,41]. Targeted next-generation sequencing (Illumina NextSeq 550 system; San Diego, CA, USA) or whole exome sequencing (Illumina NovaSeq 6000 system) were performed. Target enrichment was performed using custom-designed RNA oligonucleotide probes and a target enrichment kit (Celemics, Seoul, South Korea). Whole exome sequencing was performed using xGen Exome Research Panel v1.0 (Integrated DNA Technologies, Inc., Coraville, IA, USA) and SureSelect Human All Exon v6 enrichment kit (Agilent Technologies, Santa Clara, CA, USA).

Burrows–Wheeler Aligner software was used to align the sequence reads in the human hg19 reference genome. Single nucleotide variants and small insertions or deletions were called and crosschecked using Genome Analysis Toolkit version 3.8.0 with HaplotypeCaller and VarScan version 2.4.0. Split-read-based detection of large structural variations was conducted using Pindel and Manta. Read-depth-based detection of copy number variation was conducted using ExomeDepth version 1.1.10 [42], followed by visualization using a base-level read depth normalization algorithm designed by the authors. The variants were annotated by ANNOVAR. The 1000 Genomes Project database; Single Nucleotide Polymorphism database build 137 (dbSNP147); Genome Aggregation Database (gnomAD); the National Heart, Lung, and Blood Institute (NHLBI) Exome Sequencing Project; and the Korean Reference Genome Database were used to identify common variants. The Human Gene Mutation Database was searched to identify known pathogenic mutations. The variants were selected if they were not reported or had low frequencies (<1%) in the 1000 Genomes Project, dbSNP147, gnomAD, NHLBI Exome Sequencing Project, or Korean Reference Genome Database, and <30% heterozygous reads or <80% homozygous reads were excluded. 

The clinical importance of each variant was categorized based on the latest recommendations of the American College of Medical Genetics and Genomics standards for the interpretation and reporting of sequence variations: pathogenic, likely pathogenic, uncertain significance, benign, and likely benign variant [43,44]. We utilized the automated classification system by Intervar and Varsome, and reviewed the significance of these variants. 

## 3. Results

In this study, the demographics and clinical features of the 19 Korean patients with CSNB are documented in Table 1 and Table 2. The age at the first examination, age at symptom onset, age during the last visit, sex, SEs, initial and final BCVAs, genetic profiles, and confirmed diagnosis were presented in all 19 cases. Whether the identified genetic mutations were novel variants or previously reported is described in Table 3. The median age at initial examination was 5 years with a range of 3–21 years, the median age at symptom onset was 2 years with a range of 1–8 years, and the median age during the final visit was 11 years with a range of 5–28 years. The mean follow-up period was 5.6 ± 3.1 years. Among the 19 enrolled patients, two were male Riggs type CSNB patients, identified as harboring autosomal dominant *CNGB1* and *GNAT1* gene mutations, one was male complete type with autosomal recessive *TRPM1* gene mutation, two were male complete types with X-linked recessive *NYX* gene mutations, and 14 were male incomplete types with X-linked recessive *CACNA1F* gene mutations. The representative multimodal images of the Riggs, complete, and incomplete type patients are presented in Figure 1, Figure 2, Figure 3 and Figure 4, which include initial fundus photography, SD-OCT, and full-field standard ERG.

Two male Riggs type patients with an autosomal dominant *CNGB1* and *GNAT1* gene mutations were enrolled. The patients all presented with emmetropic features and relatively good BCVAs. The pathogenic variants were *CNGB1* [NM_001297, c.2544delG: p.(G848fs)], (NM_001297, c.1035-1G>A) and *GNAT1* [NM_000172, c.753C>A:p.(Asp251Lys)]. The complete-type patient with autosomal recessive *TRPM1* gene mutation showed high myopia with relatively good BCVAs in both eyes. The pathogenic variants were previously reported *TRPM1* [NM_002420.6, c.3280C>T:p.(Arg1094*)], [NM_002420.6, c.3794delA:p.(Asn1265Ilefs*42)] [30]. The two male complete-type patients with X-linked recessive *NYX* gene mutations showed high myopic features (mean, −7.5D) with poor BCVAs (mean, logMAR 0.7) in both eyes. One patient was too young to measure visual acuity; thus, we documented the BCVA as “poor.” The pathogenic variants were *NYX* [NM_022567.2, c.182_183insT:p.(Cys62Valfs*53)], and *NYX* (NM_022567.2, c.38-1_38delGCinsTT) mutations [45]. The patient with the novel mutation had 3–4 Hz left beating jerk-pendular nystagmus. 

The 14 male incomplete type patients with X-linked recessive *CACNA1F* gene mutations show mixed SE features (three highly myopic, seven slightly myopic, and four hyperopic cases) with poor BCVAs in both eyes maintained from the initial examination to the final visit: initial BCVA 0.76 ± 0.21 logMAR, final BCVA 0.71 ± 0.19 logMAR, median SE −1.12, and mean ± SD SE −2.4 ± 4.3 (Table 1 and Table 2). We discovered six novel pathogenic variants with one overlapping mutation: (1) NM_005183.2, exon (13–23) deletion; (2) NM_005183.2, c.2175_2179delins:p.(Gly726IIefs*61); (3) NM_005183.2, c.1910+1G>A; (4) NM_005183.2, c.4042-1G>T; (5) NM_005183.2, c.2761C>A:p.(Leu921IIe); (6) NM_005183.2, c.2767-1G>C. Seven other pathogenic variants have been described in previous literature [46,47,48,49,50,51]. Among the 14 patients with *CACNA1F* mutations, 10 (71%) had nystagmus.

In addition, we analyzed the ERG patterns of each CSNB type and summarized them in Table 2. The Riggs and *TRPM1* complete types showed low dark-adapted 3.0 ERG A-wave and B-wave amplitudes, and preserved light-adapted 3.0, and 30-Hz flicker ERG amplitude with typical electronegative ERG (B/A ratio ≤ 1) compared to age-matched normal control values: Riggs dark-adapted 3.0 A-wave amplitude 66.9 ± 8.3, B-wave amplitude 54.1 ± 4.7, B/A ratio 0.81 ± 0.05, light-adapted 3.0 A-wave amplitude 19.5 ± 2.1, B-wave amplitude 55.9 ± 3.7, and 30-Hz flicker amplitude 51.8 ± 4.9; *TRPM1* dark-adapted 3.0 A-wave amplitude 184.1 ± 38.3, B-wave amplitude 86.3 ± 1.2, B/A ratio 0.49 ± 0.09, light-adapted 3.0 A-wave amplitude 46.8 ± 4.1, B-wave amplitude 85.6 ± 0.3, and 30-Hz flicker amplitude 74.5 ± 5.1. On the other hand, the *NYX* complete and *CACNA1F* incomplete types showed substantially reduced A-wave and B-wave amplitude in dark-adapted 3.0-and light-adapted 3.0 as well as markedly decreased 30-Hz flicker ERG amplitude with electronegative ERG: *NYX* dark-adapted 3.0 A-wave amplitude 31.8 ± 6.2, B-wave amplitude 18.9 ± 1.1, B/A ratio 0.61 ± 0.08, light-adapted 3.0 A-wave amplitude 6.6 ± 0.7, B-wave amplitude 12.7 ± 3.8, and 30-Hz flicker amplitude 11.7 ± 2.6; *CACNA1F* dark-adapted 3.0 A-wave amplitude 43.7 ± 6.0, B-wave amplitude 25.3 ± 3.3, B/A ratio 0.58 ± 0.02, light-adapted 3.0 A-wave amplitude 3.8 ± 0.5, B-wave amplitude 6.5 ± 0.3, and 30-Hz flicker amplitude 5.7 ± 0.9. The characteristic “double peak” 30-Hz flicker ERG pattern (Figure 4) was observed in 9 incomplete type patients (64%).

## 4. Discussion

Our study investigated the clinical and genetic characteristics of 19 Korean patients with CSNB. We identified three types of CSNB, including Riggs and the complete and incomplete forms of Schubert–Bornschein, and their pathogenic genetic profiles. *CACNA1F* mutations appeared to be the major genetic cause in our cohort. The clinical characteristics according to CSNB type were similar to those reported in the literature; the Riggs type presented better BCVA with mild myopia compared to the Schubert–Bornschein type [3,7,52]. Among the Schubert–Bornschein type subjects, complete type patients with *TRPM1* gene mutations presented better BCVA than the others. This genotype–phenotype correlation suggests the importance of genetic analyses for predicting visual outcomes in patients with CSNB.

The pathogenic genetic mutations of the Riggs type affect the phototransduction cascade and retinoid recycling pathway [1,3]. The pathogenic genetic mutations of the complete type, *NYX* and *TRPM1*, affect glutamate-induced signaling. In the dark-adapted environment, rod photoreceptors release glutamate, which binds to GRM6 (mGluR6 receptor) and influences TRPM1 ion channel closure [53]. If the TRPM1 channel is dysregulated, constant depolarization of the ON bipolar cells leads to decreased photoreceptor sensitivity [54]. The *NYX* gene consists of two exons encoding nyctalopin and is considered to localize the TRPM1 receptor in the proper position at the dendritic tip of ON bipolar cells [18,19]. The *TRPM1* gene consists of 27 exons encoding 1642 amino acids, and the mutants displayed dysfunction of the TRPM1 channel [7,23]. The pathogenic genetic mutations of the incomplete type, *CACNA1F*, affect glutamate release, which plays a role in signal transmission from the photoreceptors to bipolar cells [55]. The *CACNA1F* gene consists of 48 exons encoding 1966 amino acids and is widely known for encoding the α1-subunit of an L-type voltage-dependent Ca^2+^ channel [56]. Dysregulation of glutamate release affects both ON and OFF bipolar cells; thus, the incomplete type is likely to present more severe visual symptoms than the complete type by influencing both rod and cone cell responses.

Clinical studies evaluating the genotype–phenotype correlation in CSNB patients have been conducted since the identification of pathogenic genes. Previous studies involving novel mutations revealed that the autosomal dominant Riggs type CSNB patients did not show typical myopia, nystagmus, strabismus, or severe visual impairment [8,9,10,11,12,13,14,15,57]. According to our case with a *CNGB1* and *GNAT1* mutation, mild visual symptoms, including good BCVAs, emmetropic to mild myopia, and night blindness, were documented, which were similar to prior clinical findings [58,59]. Electronegative ERGs with relatively preserved cone responses were also consistent with previous investigations [3]. In our cohort, compound heterozygous variants of *CNGB1* and a heterozygous variant of *GNAT1* were identified in two male Riggs type patients. Bi-allelic pathogenic variants of *CNGB1* are known to cause retinitis pigmentosa. Recently, isolated rod dysfunction associated with *CNGB1* was reported [60]. Although retinal pigmentary change may appear at a late age, we included this patient as CSNB based on the clinical features at the time of study enrollment, since all the ophthalmic examinations were completely normal with mild visual symptom of night blindness. 

There have also been numerous investigations focusing on the Schubert–Bornschein type. Miyake et al. subdivided Schubert–Bornschein type CSNB into complete and incomplete depending on the dark-adapted ERG pattern [61]. The authors found that the refractive errors were different between the complete (mostly myopic) and incomplete (variation from highly myopic to hyperopic) types. Allen et al. reported that 11 X-linked CSNB families displayed either *NYX* or *CACNA1F* gene mutations and suggested that the complete or incomplete phenotypes do not correlate with genotype, whereas some ERG features, such as oscillatory potentials and ON/OFF bipolar responses, may be helpful for indicating genotypes [62]. Bijveld et al. studied 101 Dutch patients with *NYX*, *TRPM1*, *GRM6*, and *GPR179* mutations in the complete type and *CACNA1F* and *CABP4* mutations in the incomplete type [7]. The authors pointed out that the photopic ERG pattern was the most important clinical test to distinguish between the complete and incomplete types, and complete type patients showed better visual acuity than those with the incomplete type. In addition, the incomplete type showed both rod- and cone-related visual symptoms and only half of the patients suffered night blindness compared to the complete type, which was mainly rod-related and showed 100% night blindness. The authors also found that even though there were various genetic mutations in the complete type, there was only one unique phenotype, whereas in the incomplete type, *CACNA1F* and *CABP4* mutations showed distinct visual symptoms. According to our three cases of the complete type, two cases of *NYX* gene mutations showed high myopia and poor visual acuity, whereas one case of *TRPM1* gene mutations showed high myopia but retained visual acuity. This suggests that in the complete form of the Schubert–Bornschein type, genotype confirmation may be relevant to clinical phenotypes. In our 14 cases of the incomplete type, all *CACNA1F* gene mutations presented various features of SE from high myopic to hyperopic; however, all enrolled subjects showed poor visual acuity, which is consistent with prior findings [7,37,38,60]. Moreover, according to our ERG analysis, complete *TRPM1* patients showed relatively preserved cone responses, whereas complete *NYX* and incomplete *CACNA1F* patients showed profoundly decreased cone responses, which is consistent with previous studies [3,7]. In addition, as Miyake et al. [59] reported, a 30-Hz flicker ERG double peak occurrence was detected in nine patients with incomplete ERG (64%). 

This study had some limitations. First, despite enrolling CSNB patients from two tertiary hospitals, only 19 subjects (two Riggs type, three complete types, and 14 incomplete types) were included in the study. Our study may not reflect all patients with CSNB in Korea. Due to the small study population, we could not apply the appropriate statistical genotype–phenotype correlation analysis. Additional multicenter clinical trials with larger enrollment numbers should be conducted. Moreover, potential patients with CSNB may have been missed because of diagnostic complexity. Finally, the age of symptom onset may be inaccurate in adults, and the follow-up period may not be sufficient to conclude long-term visual acuity impairment.

Nonetheless, our study has some strengths that should be mentioned. To the best of our knowledge, this is the first multicenter study to analyze and document Korean patients with CSNB as a case series. A few reports regarding novel mutations in Korea have been published; however, evaluation of the genotype–phenotype correlation in 19 Korean CSNB cases may be valuable. Moreover, we identified 13 novel mutations in this article, which are summarized in Table 3. 

## 5. Conclusions

This is the first Korean CSNB case series on the genotype–phenotype correlation. The Riggs and complete types of *TPRM1* gene mutations presented good visual acuity, whereas the incomplete and complete types of *NYX* gene mutations were frequently associated with poor visual acuity and nystagmus. Clinicians should be aware of the possibility of diagnosing CSNB when examining children with symptoms of poor visual acuity, strabismus, nystagmus, and no definite retinal structural abnormalities [4]. Moreover, genetic analysis with next-generation sequencing is recommended as a frontline diagnostic tool for CSNB patients. Future studies with larger populations should be conducted to assess the statistical genotype–phenotype correlation in Korean patients with CSNB.

## Figures and Tables

**Figure 1 genes-12-00789-f001:**
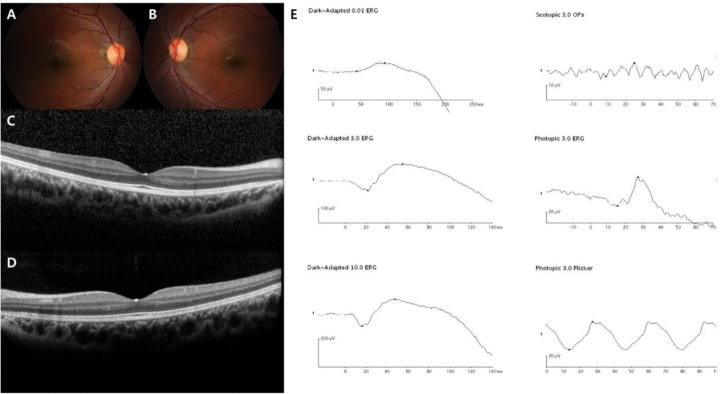
Representative images of the Riggs type CSNB patient (case 1). The patient was diagnosed with CSNB at age 9, and the last visit was at age 12. *GNAT1* was sequenced as the pathogenic gene. (**A**–**D**) Normal fundus structures observed using initial fundus photography and SD-OCT. (**E**) Full-field standard ERG showing significantly reduced dark-adapted rod responses (both the **A** and **B** waves) and normal light-adapted cone responses in the right eye.

**Figure 2 genes-12-00789-f002:**
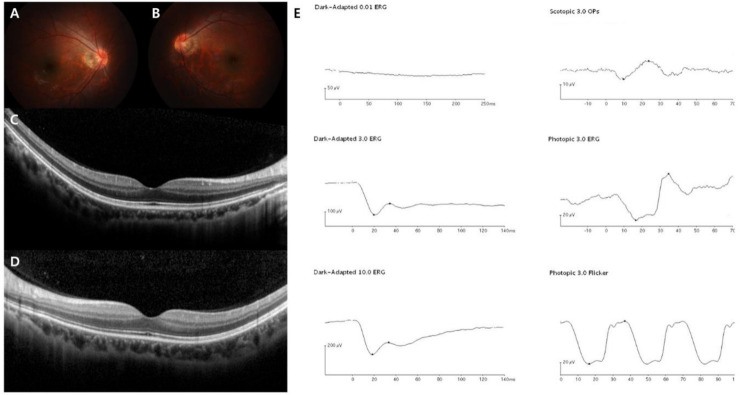
Representative images of the complete form of CSNB (case 3). The patient was diagnosed with CSNB at age 19, and the last visit was at age 28. *TRPM1* was sequenced as the pathogenic gene. (**A**–**D**) Normal fundus structures observed using initial fundus photography and SD-OCT. (**E**) Full-field standard ERG showing relatively preserved dark-adapted 3.0 A wave, while markedly reduced B wave, suggesting “electronegative ERG” pattern. Light-adapted 3.0 cone responses are slightly decreased, but normal 30-Hz flicker responses are detected in the right eye.

**Figure 3 genes-12-00789-f003:**
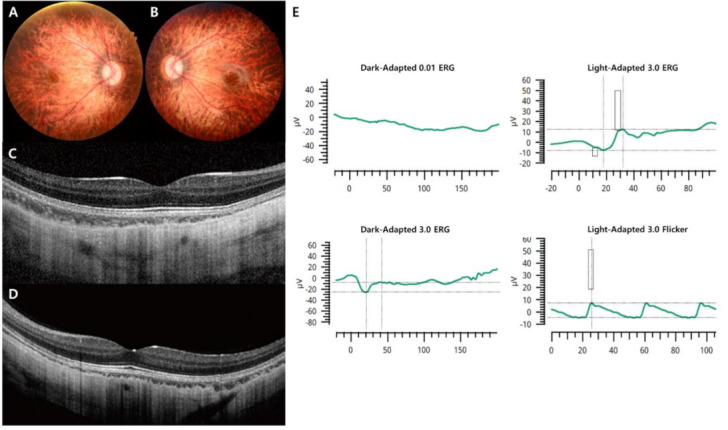
Representative images of the complete form of CSNB caused by an *NYX* mutation (case 5). (**A**,**B**) Fundus photographs showing myopic tigroid fundus. (**C**,**D**) OCT revealing normal fovea and outer retina structures but thinning of the choroidal thickness. (**E**) There was no detectable ERG in the 0.01 dark-adapted response, and electronegative ERG was observed for the 3.0 dark-adapted response. Decreased light-adapted cone responses in the right eye were detected.

**Figure 4 genes-12-00789-f004:**
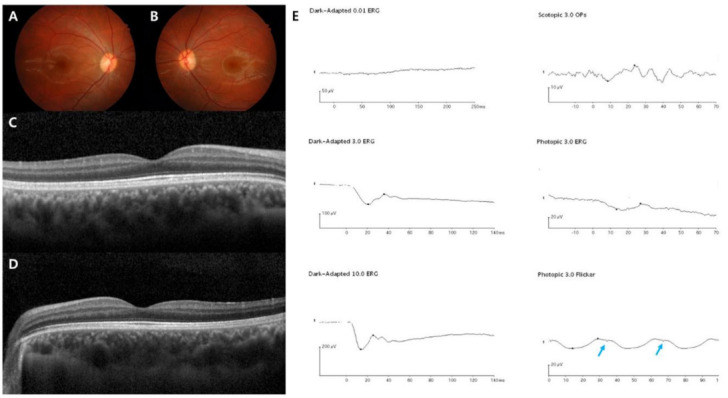
Representative images of the incomplete form of the Schubert–Bornschein type (case 16). The patient was diagnosed with CSNB at age 7, and the last visit was at age 18. *CACNA1F* was sequenced as the pathogenic gene. (**A**–**D**) Normal fundus structures observed using initial fundus photography and SD-OCT. (**E**) Full-field standard ERG showing typical electronegative ERG pattern. Unlike the complete type, significantly decreased light-adapted 3.0 cone responses and 30-Hz flicker responses were detected with a double peak sign in the right eye (blue arrow), suggesting both severe cone and rod dysfunction.

**Table 1 genes-12-00789-t001:** Summary of Clinical Characteristics of Congenital Stationary Night Blindness in 19 Patients.

No.	Sex	Age at Exam (y)	Age at Symptom Onset (y)	Age at Last Visit (y)	Diagnosis	Gene	Inheritance	SE (D)		Initial BCVA (logMAR)		Final BCVA (logMAR)		Strabismus	Nystagmus
								OD	OS	OD	OS	OD	OS		
1	M	21	1	22	Riggs	*CNGB1*	AR	−1.0	1.5	0.1	0.1	0.1	0.1	No	No
2	M	9	8	12	Riggs	*GNAT1*	AD	0.5	0.5	0.2	0.2	0.2	0.2	No	No
3 †	M	19	4	28	Complete	*TRPM1*	AR	−7.5	−6.5	0.4	0.4	0.3	0.3	No	No
4	M	3	1	7	Complete	*NYX*	XL	−6.5	−5.5	poor	poor	poor	poor	No	Yes
5	M	5	4	8	Complete	*NYX*	XL	−9.5	−9.5	0.7	0.7	0.6	0.6	Yes	No
6	M	3	1	7	Incomplete	*CACNA1F*	XL	3.0	3.0	0.7	0.8	0.7	0.7	No	Yes
7	M	3	1	8	Incomplete	*CACNA1F*	XL	−1.0	0.5	1.0	0.8	0.9	0.8	Yes	Yes
8	M	3	1	6	Incomplete	*CACNA1F*	XL	0.5	1.0	0.8	0.7	0.7	0.7	Yes	Yes
9	M	3	2	5	Incomplete	*CACNA1F*	XL	−1.0	−1.5	0.9	0.9	0.8	0.8	No	Yes
10	M	3	2	13	Incomplete	*CACNA1F*	XL	−10.5	−9.5	0.7	0.6	0.7	0.6	Yes	No
11 †	M	4	1	13	Incomplete	*CACNA1F*	XL	−11.5	−10.0	0.6	0.7	0.6	0.6	No	Yes
12	M	4	2	12	Incomplete	*CACNA1F*	XL	−3.0	−3.0	0.6	0.6	0.6	0.6	Yes	No
13	M	5	4	9	Incomplete	*CACNA1F*	XL	−1.5	−1.5	0.6	0.6	0.5	0.5	No	Yes
14	M	5	1	8	Incomplete	*CACNA1F*	XL	−2.0	−1.5	1.0	1.0	0.9	0.9	Yes	Yes
15 ‡	M	7	1	22	Incomplete	*CACNA1F*	XL	2.5	0.5	0.7	0.7	0.7	0.7	No	Yes
16	M	7	4	18	Incomplete	*CACNA1F*	XL	−9.5	−8.5	0.7	0.6	0.6	0.6	No	Yes
17	M	9	6	11	Incomplete	*CACNA1F*	XL	1.5	0.5	0.5	0.5	0.5	0.5	No	No
18	M	10	8	11	Incomplete	*CACNA1F*	XL	−1.5	−0.5	1.2	1.5	1.0	1.5	No	No
19	M	13	6	22	Incomplete	*CACNA1F*	XL	−0.5	−0.5	0.6	0.6	0.6	0.6	No	Yes
Case	Nystagmus						Case	Nystagmus						
3	3–4 Hz pendular-LBJ bilateral symmetric						13	gaze evoked nystagmus						
6	Gaze evoked nystagmus						14	3 Hz UBJ bilateral symmetric						
7	1–2 Hz RBJ-LBJ intermittent bilateral symmetric						15	4 Hz multiplanar nystagmus bilateral symmetric						
8	5 Hz small amplitude pendular nystagmus						16	gaze evoked nystagmus						
9	4 Hz fine amplitude pendular nystagmus						19	gaze evoked nystagmus						
11	3–4 Hz LBJ bilateral symmetric														

AD, autosomal dominant; AR, autosomal recessive; XL, X-linked recessive. SE, spherical equivalent; BCVA, best-corrected visual acuity; logMAR, logarithm of the minimum angle of resolution. RBJ, right beating jerk; LBJ, left beating jerk; UBJ, upbeat jerk. † Patients previously reported by our group. ‡ Genetic analysis was performed using whole exome sequencing.

**Table 2 genes-12-00789-t002:** Clinical Characteristics of Congenital Stationary Night Blindness patients based on classification types.

Type	Gene	Age at Symptom Onset (y)	Sex (M:F)	SE	Initial BCVA (logMAR)	Final BCVA (logMAR)	Strabismus (Yes:No)	Nystagmus (Yes:No)	
Riggs (*n* = 2)	*CNGB1 GNAT1*	4.5 ± 3.5	2:0	0.3 ± 0.9	0.15 ±0.05	0.15 ± 0.05	0:2	0:2	
Complete (*n* = 1)	*TRPM1*	4	1:0	−7.0	0.4	0.3	0:1	0:1	
Complete (*n* = 2)	*NYX*	2.5 ± 1.5	2:0	−7.8 ± 1.8	0.70 ± 0.00	0.60 ± 0.00	1:1 (50%)	1:1 (50%)	
Incomplete (*n* = 14)	*CACNA1F*	2.7 ± 2.2	12:0	−2.6 ± 4.4	0.79 ± 0.21	0.74 ± 0.19	5:9 (36%)	10:4 (71%)	
Type	Dark-Adapted 0.01 ERG		Dark-Adapted 3.0 ERG			Light-Adapted 3.0 ERG		30-Hz Flicker ERG	
	A-wave amplitude (μV)	B-wave amplitude (μV)	A-wave amplitude (μV)	B-wave amplitude (μV)	B/A ratio	A-wave amplitude (μV)	B-wave amplitude (μV)	Amplitude (μV)	Double peak (Yes:No)
Riggs (*n* = 2)	0.9 ± 0.2	37.2 ± 3.5	66.9 ± 8.3	54.1 ± 4.7	0.81 ± 0.05	19.5 ± 2.1	55.9 ± 3.7	51.8 ± 4.9	0:2 (0%)
Complete (*n* = 1)	0	0	184.1 ± 38.3	86.3 ± 1.2	0.49 ± 0.09	46.8 ± 4.1	85.6 ± 0.3	74.5 ± 5.1	0:1 (0%)
Complete (*n* = 2)	0	0	31.8 ± 6.2	18.9 ± 1.1	0.61 ± 0.08	6.6 ± 0.7	12.7 ± 3.8	11.7 ± 2.6	0:2 (0%)
Incomplete (*n* = 14)	3.2 ± 0.7	25.4 ± 2.8	43.7 ± 6.0	25.3 ± 3.3	0.58 ± 0.02	3.8 ± 0.5	6.5 ± 0.3	5.7 ± 0.9	9:5 (64%)
Age-matched normal controls (*n* = 30)	32.97±11.35	234.74 ± 98.65	150.02 ± 61.05	359.77 ± 89.67	2.68 ± 0.99	21.56 ± 9.67	70.40 ± 21.97	74.44 ± 21.83	0:30 (0%)

Data presented as mean ± SD. SE, spherical equivalent; BCVA, best-corrected visual acuity; logMAR, logarithm of the minimum angle of resolution.

**Table 3 genes-12-00789-t003:** Pathogenic or likely pathogenic mutations identified in 19 patients with congenital stationary night blindness.

No.	Gene	Transcript	Nucleotide Change	Amino Acid Change	Zygosity	Segregation	CADD	FATHMM	SpliceAI	MAF (Gnomad)	Domain	Novel Variant
1	*CNGB1*	NM_001297.5	c.2544delG	p.(Leu849Cysfs*15)	Hetero	-	-	0.987	-	2/249580	-	Novel
			c.1035-1G>A	-	Hetero	-	32	0.547	0.96	Not found	-	Novel
2	*GNAT1*	NM_000172.4	c.753C>A	p.(Asn251Lys)	Hetero	-	24.5	0.894	-	13/251156	α subunit of G proteins	Novel
3	*TRPM1*	NM_002420.6	c.3280C>T	p.(Arg1094*)	Hetero	-	38	0.937	-	1/249530	-	Lee et al. [30]
			c.3794delA	p.(Asn1265Ilefs*42)			26.8	0.991	-	Not found	-	
4	*NYX*	NM_022567.2	c.182_183insT	p.(Cys62Valfs*53)	Hemi	Maternal	32	0.981	-	Not found	Leucine-rich repeat	Novel
5	*NYX*	NM_022567.2	c.38-1_ 38delGCinsTT		Hemi	Maternal	14.16	-	-	Not found	-	ClinVar [45]
6	*CACNA1F*	NM_005183.2	exon(13–23) deletion		Hemi	-	-	-	-	Not found	-	Novel
7	*CACNA1F*	NM_005183.2	c.1301C>T	p.(Ala434Val)	Hemi	Maternal	17.47	0.143	0.80	Not found	-	Hove et al. [46]
8	*CACNA1F*	NM_005183.2	c.2175_2179delins27	p.(Gly726llefs*61)	Hemi	Maternal	27.3	0.999	-	Not found	Ion transport	Novel
9	*CACNA1F*	NM_005183.2	c.1910+1G>A		Hemi	-	24.9	0.976	0.95	Not found	Ion transport	Novel
10	*CACNA1F*	NM_005183.2	c.4049G>A	p.(Gly1350Asp)	Hemi	-	27.2	0.987	-	Not found	Voltage-dependent calcium channel	Kim et al. [41]
11	*CACNA1F*	NM_005183.2	c.342delC	p.(Phe115Serfs*22)	Hemi	Maternal	25.6	0.940	-	Not found	-	Rim et al. [47]
12	*CACNA1F*	NM_005183.2	c.2914C>T	p.(Arg972*)	Hemi	-	12.6	0.352	-	Not found	-	Zito et al. [48]
13	*CACNA1F*	NM_005183.2	c.4042-1G>T		Hemi	Maternal	34	0.993	0.98	Not found	Voltage-dependent calcium channel	Novel
14	*CACNA1F*	NM_005183.2	c.1910+1G>A		Hemi	-	24.9	0.976	0.95	Not found	Ion transport	Novel
15	*CACNA1F*	NM_005183.2	c.5479C>T	p.(Arg1827*)	Hemi	Maternal	36	0.202	-	1/183192	-	Wutz et al. [49]
16	*CACNA1F*	NM_005183.2	c.2576+1G>A		Hemi	-	34	0.993	0.88	1/62946	-	Wang et al. [50]
17	*CACNA1F*	NM_005183.2	c.926G>A	p.(Gly309Asp)	Hemi	-	26.2	0.974	-	Not found	Ion transport	Sun et al. [51]
18	*CACNA1F*	NM_005183.2	c.2761C>A	p.(Leu921IIe)	Hemi	Maternal	25.3	0.976	-	Not found	Ion transport	Novel
19	*CACNA1F*	NM_005183.2	c.2767-1G>C		Hemi	-	35	0.994	0.99	Not found	Ion transport	Novel

CADD, combined annotation dependent depletion; FATHMM, functional analysis through hidden Markov models; MAF, minor allele frequency.

## Data Availability

Not applicable.

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
