# Peer review of "Clinical and Genetic Characteristics of Korean Congenital Stationary Night Blindness Patients"

_genes, 2021, doi:10.3390/genes12060789_

Round 1
Reviewer 1 Report
In this study, authors investigated phenotypic and genotypic correlations of 19 Korean patients with CSNB.
Introduction: line 50, what do you mean by "genetic mutations in the family pedigree "?
lines 51-69: there is a repeat in the ERG data already presented in the previous paragraph,
Figures: you presented for each figure one fundus photo and OCT of each eye but in the legend, you spoke about initial and final fundus photography and SD-OCT at follow-up, please correct it
Figures 1 and 2: ERG figure is unreadable
in figure 4, please present the double peak sign on ERG with an arrow
What about Multifocal ERG and Goldmann perimetry results ?
Was the follow-up of the patients also done by full-field ERG ? Did you notice a significant difference depending on the type of CSNB?
Author Response
In this study, authors investigated phenotypic and genotypic correlations of 19 Korean patients with CSNB.
- Introduction: line 50, what do you mean by "genetic mutations in the family pedigree "?
-> Thank you for your comment. We mistakenly wrote “in the family pedigree.” We have erased the phrase, and the sentence is now “The Riggs type genetic mutations have been previously reported…”
- lines 51-69: there is a repeat in the ERG data already presented in the previous paragraph,
-> Thank you for your suggestion. In the previous paragraph, lines 46–47, we have omitted the ERG information and placed it at lines 55–58 to avoid repeating sentences.
- Figures: you presented for each figure one fundus photo and OCT of each eye but in the legend, you spoke about initial and final fundus photography and SD-OCT at follow-up, please correct it.
-> Thank you for your comment. We have changed the figure legend as “initial fundus photography and SD-OCT” and erased final word.
- Figures 1 and 2: ERG figure is unreadable.
-> Thank you for your suggestion. We have enlarged the ERG figure to achieve better resolution.
- In figure 4, please present the double peak sign on ERG with an arrow
-> We have now included a blue arrow in Figure 4 to present the double peak sign.
- What about Multifocal ERG and Goldmann perimetry results ?
-> Cases representing Figures 1 to 4 all involved normal multifocal ERGs and Goldmann perimetry visual fields. First, we included the examinations; however, before submitting the manuscript draft, we omitted these examinations to present enlarged ERG figures (we thought that there might not have been enough space to include both multifocal ERG and Goldmann perimetry data).
- Was the follow-up of the patients also done by full-field ERG ? Did you notice a significant difference depending on the type of CSNB?
-> The follow-up of full-field ERG was performed in most patients, and there were no significant differences depending on the type of CSNB. We did not present the final follow-up ERG data in the figures because there were no definite changes in the ERG parameters.
Reviewer 2 Report
The paper by Kim, Joo, Han and Woo entitled “Clinical and genetic characteristics of Korean congenital stationary night blindness patients” describes the clinical and genetical characteristics of 19 male Korean patients with CSNB. The description of the patients include their genetic profiles, best-corrected visual acuity, the presence or absence of strabismus and nystagmus, ERGs, fundus analysis and SD-OCT. The novel aspect of this study is the first description of a Korean patient group with CSNB. Furthermore, the authors found 13/14 novel mutations in those patients and therefore add to a bigger international picture of CSNB.
Major:
- The authors mention in the manuscript that the oscillation frequency of the eyes of the NXY patient had a oscillation frequency of 3-4 Hz and describe the waveform of the nystagmus (#189). This information is missing for the patients with incomplete CSNB with nystagmus. Can the authors add this information per patient in the table?
- Could the authors specify how the frequency of the nystagmus was determined?
Minor comments
- #60 “show moderate visual symptoms” instead of “present with moderate visual symptoms”
- #90/170 Typo: spelling of Schubert-Bornschein wrong (h missing)
- #135 “an” needs to be removed (mutations is plural) “as harboring an autosomal dominant 135 CNGB1 and GNAT1 gene mutations”
- #130 The information of the age of the patient includes median and mean- one would be sufficient “The median age at initial examination was 5 years with a range of 3–21 years (mean ± SD, 7.1 ± 5.2 years), the median age at symptom onset was 2 years with a range of 1–8 years (3.1 ± 2.4 years), and the median age during the final visit was 11 years with a range of 5–28 years (12.7 ± 6.4 133 years).”
- For table 1 it could be nice to highlight changes in the BCVA between the initial and final assessment to give a quicker/clearer overview)
- The headers of table 2 are improperly formatted
- The layout of table 3 is not clearly arranged
- The style of ERG figures in figure 1,2 and 4 are different from 3. It would be good to keep them consistent. Furthermore, figure 1,2 and 4 have a bad resolution (at least on my computer).
- #244 Suggestion: “affects” instead of “leads” (“Dysregulation of glutamate release leads to both ON and OFF bipolar cells”)
- #307 The authors claim that they found 13 novel mutations, but according to table 3 it should be 14.
Author Response
Major
- The authors mention in the manuscript that the oscillation frequency of the eyes of the NXY patient had a oscillation frequency of 3-4 Hz and describe the waveform of the nystagmus (#189). This information is missing for the patients with incomplete CSNB with nystagmus. Can the authors add this information per patient in the table?
-> Thank you for your suggestion. We have added the nystagmus patterns in Table 1.
- Could the authors specify how the frequency of the nystagmus was determined?
-> The frequency of nystagmus was determined by visual inspection in most patients, except in patients 7 and 11. Video nystagmography was performed only in these two patients. Therefore, we thought that it was more reasonable to state that the type and frequency of nystagmus was determined by visual inspection.
Minor
- #60 “show moderate visual symptoms” instead of “present with moderate visual symptoms”
-> Thank you for your comment. We have corrected this phrase as suggested.
- #90/170 Typo: spelling of Schubert-Bornschein wrong (h missing)
-> We have corrected the typographical error. Thank you for your comment.
- #135 “an” needs to be removed (mutations is plural) “as harboring an autosomal dominant 135 CNGB1 and GNAT1 gene mutations”
-> We have removed “an” due to the plural terminology. Thank you for your advice.
- #130 The information of the age of the patient includes median and mean- one would be sufficient “The median age at initial examination was 5 years with a range of 3–21 years (mean ± SD, 7.1 ± 5.2 years), the median age at symptom onset was 2 years with a range of 1–8 years (3.1 ± 2.4 years), and the median age during the final visit was 11 years with a range of 5–28 years (12.7 ± 6.4 133 years).”
-> Thank you for your comment. As suggested, we have deleted the mean age data.
- For table 1 it could be nice to highlight changes in the BCVA between the initial and final assessment to give a quicker/clearer overview)
-> Thank you for your suggestion. We have marked the changes in the BCVA between the initial and final follow-ups as bold characters.
- The headers of table 2 are improperly formatted.
-> Thank you for your comment. In the manuscript draft as submitted to Genes, we formatted the Table 2 headers correctly. We believe that the journal editors will format the Table section properly and clearly in the final draft.
- The layout of table 3 is not clearly arranged.
-> Thank you for your comment. In the manuscript draft as submitted to Genes, we edited the layout clearly. However, the table layout might be changed in the peer-review process to fit the journal style. I reckon that the journal editors will format the Table section clearly in the final draft.
- The style of ERG figures in figure 1,2 and 4 are different from 3. It would be good to keep them consistent. Furthermore, figure 1,2 and 4 have a bad resolution (at least on my computer).
-> Thank you for your suggestion. We have enlarged the ERG figures to achieve better resolution. Figures 1, 2, and 4 are from one tertiary hospital, and Figure 3 is from another tertiary hospital (our study included two tertiary hospitals). Therefore, the styles of the ERG figures are different.
- #244 Suggestion: “affects” instead of “leads” (“Dysregulation of glutamate release leads to both ON and OFF bipolar cells”)
-> Thank you for your comment. We have changed “leads to” to “affects.”
- #307 The authors claim that they found 13 novel mutations, but according to table 3 it should be 14.
-> Thank you for your comment. As presented in Table 3, there are 14 novel variants, but cases 9 and 14 both involved c.1910+1G>A novel variants as a duplication. Therefore, we have addressed that we found 13 novel mutations.
Reviewer 3 Report
The authors investigated the clinical and genetic findings of 19 Korean patients with congenital stationary night blindness (CSNB). Clinically, funduscopy, OCT and electroretinography (ERG) were performed in all patients. In addition, genetic testing using gene targeted panel and whole exome sequencing were performed to identify mutations. This study was the largest case series of Korean patients with CSNB. I agree that the evaluation of genotype-phenotype correlation of Korean CSNB patients was valuable. However, I found some concerns about evaluation of ERG data.
1. In Table 2, the ERG amplitudes should be compared with age-matched normal controls. The two patients with NYX mutations exhibited extremely reduced a-waves of dark-adapted 3.0 ERG. ERG amplitudes data of age-matched normal controls should be shown in DA 3.0, LA 3.0 and 30-Hz flicker.
2. In Table 2, regarding ERG, the authors mentioned and performed ERG recording according to the ISCEV protocol in the methods section. I did not find any dark-adapted 0.01 ERG data in Table 2. Dark-adapted 0.01 ERG data should be shown, otherwise complete CSNB cannot be differentiated from incomplete CSNB.
3. In Table 3, what do bold letters mean in CNGB1?
4. In Table 3, how many GNAT1 mutations were found in patient #2 with autosomal dominant inheritance?
5. In Table 3, 5 missense mutations were found. Are these mutations really pathogenic according to the American College of Medical Genetics and Genomics (ACMG) guideline. Also, I wonder the authors performed the segregation analysis using DNA samples of each proband's relatives, especially in families with the missense mutations.
6. Figure 1, light-adapted cone responses were markedly reduced, in addition to dark-adapted ERG. I think that this evaluation was correct. Generally, these ERG forms were called "subnormal ERG". The features of Riggs type CSNB are normal light-adapted cone responses as mentioned the following two important references, which should be cited in this manuscript. The diagnosis of patient 1 with the CNGB1 mutation is not "Riggs type" CSNB.
- Riggs-type dominant congenital stationary night blindness: ERG findings, a new GNAT1 mutation and a systemic association. Marmor MF, Zeitz C. Doc Ophthalmol. 2018 Aug;137(1):57-62.
- A Novel Heterozygous Missense Mutation in GNAT1 Leads to Autosomal Dominant Riggs Type of Congenital Stationary Night Blindness. Zeitz C, Méjécase C, Stévenard M, Michiels C, Audo I, Marmor MF. Biomed Res Int. 2018 Apr 23;2018:7694801.
7. As above mentioned, the authors may misunderstand the diagnosis of "Riggs type" CSNB. Therefore, I strongly recommend that the ISCEV ERG raw data of patient 2 with GNAT1 mutation should be shown as a new figure to clarify whether the ERG waveforms represent ERG data of "Riggs type" CSNB.
8. In Figure 4, the light-adapted cone responses look like extinguished, not reduced. The authors should discuss why the light-adapted cone responses were extinguished in complete CSNB. Overall, description of ERG data should be accurately mentioned in the Results and Figure legends sections.
Author Response
- In Table 2, the ERG amplitudes should be compared with age-matched normal controls. The two patients with NYX mutations exhibited extremely reduced a-waves of dark-adapted 3.0 ERG. ERG amplitudes data of age-matched normal controls should be shown in DA 3.0, LA 3.0 and 30-Hz flicker.
-> Thank you for your suggestion. Age-matched normal control data are now presented in Table 2.
- In Table 2, regarding ERG, the authors mentioned and performed ERG recording according to the ISCEV protocol in the methods section. I did not find any dark-adapted 0.01 ERG data in Table 2. Dark-adapted 0.01 ERG data should be shown, otherwise complete CSNB cannot be differentiated from incomplete CSNB.
-> Thank you for your suggestion. We have added dark-adapted 0.01 ERG data in Table 2.
- In Table 3, what do bold letters mean in CNGB1?
-> Thank you for your comment. In the manuscript draft we submitted, there were no bold letters in CNGB1. We think that during the process of editing in accordance with journal style, bold letters might have been accidentally included. We have changed the bold lettering to plain style.
- In Table 3, how many GNAT1 mutations were found in patient #2 with autosomal dominant inheritance?
-> One GNAT1 mutation was found in patient #2 with autosomal dominant inheritance.
- In Table 3, 5 missense mutations were found. Are these mutations really pathogenic according to the American College of Medical Genetics and Genomics (ACMG) guideline. Also, I wonder the authors performed the segregation analysis using DNA samples of each proband's relatives, especially in families with the missense mutations.
-> We believe that the mutations were pathogenic according to the ACMG guideline. We performed the segregation analysis using DNA samples from each proband's relatives.
- Figure 1, light-adapted cone responses were markedly reduced, in addition to dark-adapted ERG. I think that this evaluation was correct. Generally, these ERG forms were called "subnormal ERG". The features of Riggs type CSNB are normal light-adapted cone responses as mentioned the following two important references, which should be cited in this manuscript. The diagnosis of patient 1 with the CNGB1 mutation is not "Riggs type" CSNB.
Riggs-type dominant congenital stationary night blindness: ERG findings, a new GNAT1 mutation and a systemic association. Marmor MF, Zeitz C. Doc Ophthalmol. 2018 Aug;137(1):57-62.
A Novel Heterozygous Missense Mutation in GNAT1 Leads to Autosomal Dominant Riggs Type of Congenital Stationary Night Blindness. Zeitz C, Méjécase C, Stévenard M, Michiels C, Audo I, Marmor MF. Biomed Res Int. 2018 Apr 23;2018:7694801.
--> Thank you for your comment. We have added the two citations in the Discussion section (Page 9, line 248). We have added the control ERG data in Table 2. In our study, both CNGB1 and GNAT1 appeared to present normal light-adapted cone responses compared to the normal control data. The normal control light-adapted 3.0 ERG A-wave amplitude was 21.56±9.67 μV and B-wave amplitude was 70.40±21.97 μV. The CNGB1 mutation light-adapted 3.0 A-wave amplitude was 21.6 μV (98th percentile) and B-wave amplitude was 52.2 μV (65th percentile); the GNAT1 mutation light-adapted 3.0 A-wave amplitude was 17.4 μV (76th percentile) and B-wave amplitude was 59.6 μV (72nd percentile). These were all in the normal range of age-matched control data. As we reviewed the red ERG waveforms of the normal reference, we found that they were slightly different from our age-matched average control data. Thus, we deleted the normal reference waveforms in the figures to avoid confusion. Moreover, as suggested, previously published articles reported that the Riggs-type CSNB presents low dark-adapted 3.0 ERG A wave and B wave and normal cone responses, which agree with our CNGB1 and GNAT1 cases. Therefore, we decided to retain the two cases caused by CNGB1 and GNAT1 mutations as Riggs-type CSNB.
- As above mentioned, the authors may misunderstand the diagnosis of "Riggs type" CSNB. Therefore, I strongly recommend that the ISCEV ERG raw data of patient 2 with GNAT1 mutation should be shown as a new figure to clarify whether the ERG waveforms represent ERG data of "Riggs type" CSNB.
-> Thank you for your valuable comment. As suggested, we have replaced Figure 1 from CNGB1 mutation case to GNAT1 mutation case. Both CNGB1 and GNAT1 patients showed normal light-adapted cone responses in accordance with our age-matched control values. Thus, we believe that both mutations are compatible with Riggs type CSNB.
- In Figure 4, the light-adapted cone responses look like extinguished, not reduced. The authors should discuss why the light-adapted cone responses were extinguished in complete CSNB. Overall, description of ERG data should be accurately mentioned in the Results and Figure legends sections.
-> Thank you for your comment. Figure 4 is incomplete type CSNB, and thus light-adapted cone responses were almost extinguished in ERG findings.
Round 2
Reviewer 3 Report
The revised version of the manuscript was improve appropriately. I have no further comment.